# Cyano Enone-Bearing Triterpenoid Soloxolone Methyl Inhibits Epithelial-Mesenchymal Transition of Human Lung Adenocarcinoma Cells In Vitro and Metastasis of Murine Melanoma In Vivo

**DOI:** 10.3390/molecules25245925

**Published:** 2020-12-14

**Authors:** Andrey V. Markov, Kirill V. Odarenko, Aleksandra V. Sen’kova, Oksana V. Salomatina, Nariman F. Salakhutdinov, Marina A. Zenkova

**Affiliations:** 1Institute of Chemical Biology and Fundamental Medicine, Siberian Branch of the Russian Academy of Sciences, Lavrent’ev ave., 8, 630090 Novosibirsk, Russia; k.odarenko@yandex.ru (K.V.O.); alsenko@mail.ru (A.V.S.); ana@nioch.nsc.ru (O.V.S.); marzen@niboch.nsc.ru (M.A.Z.); 2N.N. Vorozhtsov Novosibirsk Institute of Organic Chemistry, Siberian Branch of the Russian Academy of Sciences, Lavrent’ev ave., 9, 630090 Novosibirsk, Russia; anvar@nioch.nsc.ru

**Keywords:** epithelial-mesenchymal transition, tumor, metastasis, triterpenoids, natural compounds, soloxolone methyl, CDDO-Me, motility, molecular docking, cytoscape

## Abstract

Introduction of α-cyano α,β-unsaturated carbonyl moiety into natural cyclic compounds markedly improves their bioactivities, including inhibitory potential against tumor growth and metastasis. Previously, we showed that cyano enone-bearing derivatives of 18βH-glycyrrhetinic (GA) and deoxycholic acids displayed marked cytotoxicity in different tumor cell lines. Moreover, GA derivative soloxolone methyl (SM) was found to induce ER stress and apoptosis in tumor cells in vitro and inhibit growth of carcinoma Krebs-2 in vivo. In this work, we studied the effects of these compounds used in non-toxic dosage on the processes associated with metastatic potential of tumor cells. Performed screening revealed SM as a hit compound, which inhibits motility of murine melanoma B16 and human lung adenocarcinoma A549 cells and significantly suppresses colony formation of A549 cells. Further study showed that SM effectively blocked transforming growth factor β (TGF-β)-induced epithelial-mesenchymal transition (EMT) of A549 cells: namely, inhibited TGF-β-stimulated motility and invasion of tumor cells as well as loss of their epithelial characteristics, such as, an acquisition of spindle-like phenotype, up- and down-regulation of mesenchymal (vimentin, fibronectin) and epithelial (E-cadherin, zona occludens-1 (ZO-1)) markers, respectively. Network pharmacology analysis with subsequent verification by molecular modeling revealed that matrix metalloproteinases MMP-2/-9 and c-Jun N-terminal protein kinase 1 (JNK1) can be considered as hypothetical primary targets of SM, mediating its marked anti-EMT activity. The inhibitory effect of SM on EMT revealed in vitro was further confirmed in a metastatic model of murine B16 melanoma: SM was found to effectively block metastatic dissemination of melanoma B16 cells in vivo, increase expression of E-cadherin and suppress expression of MMP-9 in lung metastatic foci. Altogether, our data provided valuable information for a better understanding of the antitumor activity of cyano enone-bearing semisynthetic compounds and revealed SM as a promising anti-metastatic drug candidate.

## 1. Introduction

Epithelial-mesenchymal transition (EMT) is a reversible process during which tumor cells lose their apical-basal polarity and contacts with adjacent cells, as well as acquire motile and invasive behavior. EMT is one of the key factors exerting a marked stimulation of tumor progression and metastasis, and weakening the sensitivity of tumor cells to chemo- and immunotherapy [1]. The acquisition of a mesenchymal phenotype by malignant cells via EMT and their subsequent dissemination and outgrowth at distant organs was shown not only for a wide range of epithelial tumors, including lung, breast, pancreatic, prostate, liver and other carcinomas [2], but also for melanoma [3] and hematological malignancies [4].

EMT of tumor cells can be induced by different stimulus from tumor microenvironment, primarily the stromal cells, surrounding the malignant tissue, e.g., myeloid-derived suppressor cells, cancer-associated fibroblasts and macrophages [5]. These cells can secrete an array of growth factors and cytokines, such as transforming growth factor β (TGF-β), epidermal growth factor (EGF), vascular endothelial growth factor (VEGF), interleukin 6 (IL-6) and others, which bind to the corresponding receptors exposed on the surface of tumor cells and thereby activate a complex network of signaling pathways with subsequent up-regulation of EMT-associated transcription factors ZEB1/2, Snail, Slug and Twist1/2 that orchestrate EMT program [1,5]. Upon the activation of EMT, the expression of epithelial cadherin (E-cadherin), being the major component of the cell-cell adherens junctions [6], is markedly repressed, which causes the loss of epithelial cobblestone-like morphology. Besides this, the events occurring during EMT also include the rearrangement of cytoskeleton (e.g., a switch from cytokeratins to vimentin), the acquisition of a spindle-shaped mesenchymal phenotype and the up-regulation of mesenchymal markers, such as neural cadherin (N-cadherin), fibronectin and vimentin [1,2,5]. As a result of these changes, the tumor cells lose their contacts with neighbor cells and basement membrane, acquire highly invasive properties and can disseminate to distant organs [5]. Thus, EMT plays a key role in malignant progression and increase of severity of metastasis. In accordance with this, a range of correlation studies showed that EMT-associated markers can be considered as powerful prognostic factors in different types of tumors, which are associated with poor prognosis [7,8,9].

Considering the tight linkage of EMT with tumor progression, extensive efforts have been dedicated to search and development of novel agents, being able to control this process in malignant cells. One of the important sources of such compounds are natural metabolites. According to the recent reviews of Feng et al. and Avila-Carrasco et al., compounds from natural products, including terpens, flavonoids, steroids, coumarins, etc., can effectively inhibit EMT of a variety of tumor cells by affecting different EMT-associated signaling pathways [10,11]. To date, a range of evidences have been collected that pentacyclic triterpenoids can effectively inhibit cellular migration and invasion by blocking EMT process [12,13,14,15,16,17,18,19,20,21]. Of note, as far as we know, these studies were mainly concerned with the investigation of natural triterpenoids [13,14,15,17,22] and triterpenoid saponins [23,24], whereas their semisynthetic analogs, generally showing a more promising bioactivity compared to natural compounds, are poorly investigated in this field. Furthermore, it is still not fully understood which master regulators control the response of mesenchymal-like tumor cells on these compounds. According to published data, only indirect effects of triterpenoids on the activation of key EMT-associated signaling pathways [12,13,17,18,20,21,22,23] and the expression of metastasis-related metadherin [16] and redox sensitive protein HMGB1 [24] were shown in this connection. Only Wang et al. reported recently that oleanolic acid can inhibit EMT of hepatocellular carcinoma cells by direct interaction with inducible form of nitric oxide (II) synthase (iNOS) [14]. Thus, further advancement of knowledge about both novel triterpenoid modulators of EMT and their molecular mechanisms of action is highly important for the development of new generation of antitumor agents.

It is know that chemical transformation of natural compounds is a promising strategy to improve their biological activities. In the case of triterpenoids, the introduction of cyano enone pharmacophore group into their structure was found to significantly reinforce the antitumor potential of parent molecules: it was shown that cyano enone-bearing triterpenoids (CETs) (Figure 1) effectively inhibited the proliferation of tumor cells, their motility and invasiveness in vitro and tumor growth and metastasis in a range of murine models in vivo [25]. Along with marked antitumor effects, these compounds display a wide range of other bioactivities, including inhibition of inflammation [26] and influenza A virus reproduction [27], immunomodulation [28] and cytoprotection [29]. Moreover, several clinical trials have been conducted to analyze the efficacy of cyano enone-bearing derivative of oleanolic acid CDDO-Me (also known as bardoxolone methyl) in patients with advanced solid tumors and lymphoid malignancies (Phase 1) [30], chronic kidney disease (Phase 3) [31], pulmonary hypertension (Phase 3) [32], Alport Syndrome (Phase 2/3) [33] and COVID-19 (Phase 2/3) [34]. Despite the proven high bioactivity of CETs, the influence of these compounds on EMT of tumor cells has been poorly investigated. Only Wang et al. reported that CDDO-Me effectively blocked this process in esophageal squamous carcinoma cells [20], however the effects of CETs on EMT of other tumor cell types have not yet been published.

Previously, our group synthesized and characterized soloxolone methyl (SM) and trioxolone methyl (TM) (Figure 1), cyano enone-bearing derivatives of 18βH-glycyrrhetinic acid, being structural analogs of CDDO-Me [35,36]. We showed that SM induced tumor cell death by triggering of mitochondrial apoptosis [35] and the stress of endoplasmic reticulum [37] and effectively inhibited the growth of murine Krebs-2 carcinoma in vivo [38], whereas TM significantly suppressed dextran sulphate sodium-induced colitis in mice [36]. In this work, we questioned whether SM and TM can modulate EMT of tumor cells. Furthermore, given the fact that introduction of cyano enone pharmacophore caused the marked strengthening of antitumor activity not only on triterpenoid platform but also on other polycyclic systems [39,40], the list of analyzed compounds was extended with bioactive cyano enone-bearing derivative of deoxycholic acid pi-153 and pi-156 (Figure 1), which, according to our earlier work [41], display the moderate cytotoxicity with respect to tumor cells. Here, we reported that SM showed the most pronounced inhibitory effect on a motility of murine melanoma B16 and human lung adenocarcinoma A549 cells compared to other investigated cyano enone-containing compounds and effectively blocked TGF-β-induced EMT of A549 cells, by inhibiting of a range of events triggered by this growth factor. Revealed anti-EMT activity of SM was further validated in metastatic model of murine B16 melanoma. Computer modeling studies were also performed to find probable primary targets of SM, associated with its anti-EMT activity.

## 2. Results

### 2.1. Screening of Cyano Enone-Bearing Compounds to Inhibit Motility of Tumor Cells

Given the high antitumor potential of cyano enone-containing polycyclic molecules and a poor understanding of their modulatory effects on EMT of tumor cells [25,39,40], we selected for the study four compounds, bearing cyano enone pharmacophore, investigated by us previously [35,41]: two derivatives of 18βH-glycyrrhetinic acid (SM and TM) and two derivatives of deoxycholic acid (pi-153 and pi-156) (Figure 1). On the first step of the study the cytotoxicity of mentioned compounds was studied in murine melanoma B16 and human lung adenocarcinoma A549 cells. As depicted in Figure 2, cytotoxicity of semisynthetic triterpenoids was significantly higher than that of deoxycholic acid derivatives: marked decrease of cell viability was observed in cells treated with 1 µM of SM and TM, whereas, in the case of pi-153 and pi-156, similar effects were detected at 100-times higher concentration of the compounds (Figure 2). Considering the fact that investigated triterpenoids (SM, TM) and steroids (pi-153, pi-156) did not show any significant cytocidal effects in concentrations up to 0.5 µM and 50 µM, respectively, mentioned dosages were further chosen for subsequent experiments to exclude the influence of anti-proliferative effects of the compounds on their probable anti-EMT activity.

It is known that an acquisition of mesenchymal phenotype by tumor cells is mainly accompanied by significant increase of cell motility [5] and low molecular weight inhibitors of EMT can effectively block the migration of cells both in the presence or absence of EMT stimulatory factors [10]. Thus, in order to evaluate a probable inhibitory potential of the cyano enone-bearing compounds against highly-motile cellular phenotype, we analyzed their effects on migration capacity of tumor B16 and A549 cells using a scratch assay. Obtained results showed that only SM inhibited wound closure rate of both cell lines (Figure 3). It was found that treatment of the cells by TM and pi-156 had no effect on their motility, whereas pi-153 suppressed the migration activity of only A549 cells. Based on the revealed ability of SM to inhibit cellular motility of both tumor cell lines, this triterpenoid was identified as a hit compound and was chosen for further investigations. Evaluation of its cytotoxicity in non-transforming human gingival fibroblasts (HGF) showed that SM is characterized by moderate selectivity toward tumor cells, displaying IC_50_^HGF^ (the concentration of SM required to inhibit 50% cell growth) = 5.9 ± 0.4 µM (Appendix A) versus IC_50_^A549^ = 2.2 ± 0.1 µM (24 h treatment period; the latter value was reported in our earlier study [27]). In spite of this, obtained results clearly demonstrated that selected working concentration of SM (0.5 µM) is non-toxic in both malignant and non-malignant cells.

Next, in order to evaluate a probable anti-metastatic potential of SM, we analyzed its inhibitory effect on the ability of single tumor cells to growth into a colony. Performed experiments revealed that SM at nanomolar concentrations effectively and dose-dependently decreased colony forming capacity of A549 cells by 2.5–10 fold compared to the control (Figure 3C). These results clearly demonstrate the expediency of further investigation of SM as the probable modulator of EMT of tumor cells.

### 2.2. Identification of a Working Concentration of TGF-β

As noted above, EMT of tumor cells can be induced by a wide range of external stimuli, among which transforming growth factor β (TGF-β) is the most known and widely reported in literature [42]. Given this fact, TGF-β was selected as a stimulator of EMT for further studies. In order to reveal the working concentration of TGF-β, we evaluated the effects of this protein on three independent EMT-associated characteristics of tumor cells, such as an endowing the cells with mesenchymal-like morphology traits, their increased migration capacity and the expression of EMT-related markers (Figure 4).

Obtained results demonstrated that TGF-β at concentrations of 10 and 20 ng/mL stimulated the acquisition by A549 cells of a spindle-shaped morphology in a dose-dependent manner (Figure 4A). Subsequent analysis of the effect of these concentrations of TGF-β on the motility of A549 cells revealed that only 20 ng/mL of TGF-β significantly reinforced this process at 24 h after treatment, whereas at time point 48 h this effect disappeared (Figure 4B). Increasing of TGF-β dosage up to 50 ng/mL was found to stimulate cell migration at both 24 h and 48 h after treatment (Figure 4B). Considering this results, two mentioned concentrations of TGF-β (20 and 50 ng/mL) were further tested for the ability to affect the expression of E-cadherin and vimentin, being known epithelial and mesenchymal markers, respectively. Performed RT-PCR analysis showed that TGF-β effectively suppressed expression of E-cadherin and increased expression of vimentin in a dose-dependent manner: it was found that treatment of A549 cells by TGF-β at 20 and 50 ng/mL for 48 h led to a down-regulation of E-cadherin mRNA by 3.2- and 9.4-fold and an up-regulation of vimentin mRNA by 2- and 3.1-fold compared to the control, respectively (Figure 4C). Thus, obtained results clearly demonstrated that TGF-β indeed induced EMT of A549 cells and 50 ng/mL of TGF-β, causing the more pronounced EMT-associated changes, was chosen as its working concentration for further studies.

### 2.3. SM Effectively Inhibits TGF-β-Driven EMT of Human Lung Adenocarcinoma A549 Cells

#### 2.3.1. SM Inhibits an Acquisition of Spindle-Like Morphology by TGF-β-Treated A549 Cells

On the next step of the study the ability of SM to affect EMT of A549 cells, induced by TGF-β, was investigated. Firstly, we questioned whether SM inhibited morphological changes in A549 cells, stimulated by mentioned growth factor. To understand this, the cells were incubated in the presence or absence of TGF-β and SM for 48 h followed by the evaluation of cellular morphology by phase-contrast microscopy. As shown in Figure 5A, control cells were characterized by low level of cellular elongation-approximately 61% of the cells in the control group had the roundish shape with the spindle-like cell morphology (SLCM) score less than 3, i.e., a ratio of their long axis to short axis on the obtained photographs did not exceeded 3.

The activation of A549 cells by TGF-β caused serious perturbation of cellular morphology parameters. The TGF-β-treated cells had significantly more elongated phenotype compared to control cells; marked depletion of rounded cell population (SLCM score <3; 22% versus 61% in the control) and an appearance of spindle-like cells (SLCM score > 5; 63% versus 18% in the control) were observed in this group (Figure 5A). The treatment of TGF-β-stimulated A549 cells by SM shifted their morphological profile towards roundish-shaped cells with low SLCM score (< 4) up to 50% versus 37% in TGF-β-treated cells and depleted the group of spindle-like cells with the highest SLCM score >7 by approximately 2-fold compared to TGF-β-activated cells (Figure 5A). Furthermore, observed effect of SM is statistically significant (*p* = 0.0028) (Appendix A). As depicted in the diagram in Figure 5A, the treatment of A549 cells by SM alone led to moderate elongation of the cells in comparison to the control, which clearly demonstrated the absence of toxic effect of this triterpenoid on the cells at used concentration because of rounding is a well-known marker of dying cells [43]. Thus, obtained results showed that SM markedly inhibited TGF-β-stimulated acquisition of mesenchymal-like phenotype by A549 cells.

#### 2.3.2. SM Inhibited Migration and Invasion of TGF-β-Stimulated A549 Cells

Given the fact that mesenchymal-like cells are characterized by high motile and invasive behavior [1,2,5], we investigated the ability of SM to block these characteristics in A549 cells undergoing TGF-β-induced EMT. Firstly, we studied the effect of SM on the motility of malignant cells using the scratch assay. Obtained results demonstrated that the incubation of TGF-β-stimulated cells in the presence of SM significantly decreased their motility by 1.4-fold at 24 h and 48 h (*p* < 0.05) compared to untreated TGF-β-stimulated cells (Figure 5B).

In order to analyze observed inhibitory effect of SM on the migration of A549 cells in real-time mode, electrical impedance assay technology (xCELLigence, ACEA Biosciences, USA) was applied. A549 cells were seeded in an upper chamber of a CIM-Plate in the presence or absence of TGF-β and SM and the level of their migration to a lower chamber, containing 10% fetal bovine serum (FBS), was measured by evaluation of the impedance of sensor electrodes mounted at the lower side of porous membrane separating the upper and lower chambers of the plate. As shown in Figure 5C, TGF-β significantly increased the motility of tumor cells: the cells, treated by this EMT stimulator, were characterized by 4.3- and 1.4-fold higher cell index in comparison with control cells at 24 h and 48 h, respectively. Incubation of TGF-β-simulated A549 cells in the presence of SM effectively inhibited their motility up to the level of control untreated cells (Figure 5C). Interestingly, that SM alone did not affect transwell migration of unstimulated A549 cells (Figure 5C), whereas in our scratch assay, described above, the treatment of these cells by SM significantly decreased their wound closure rate (Figure 3B). We suppose that this discrepancy can be explained by the presence of chemoattractant (10% FBS) in the lower chambers of the CIM-plate, which can outweigh the inhibitory effect of SM on the basal level of cellular motility. Nevertheless, the data obtained from two independent experiments clearly showed that SM effectively blocked the EMT-associated acquisition of highly motile phenotype by lung adenocarcinoma A549 cells.

Next, we questioned whether SM can modulate the invasion capacity of TGF-β-stimulated tumor cells. To understand this, we repeated the analysis of cellular motility on xCELLigence platform by using the CIM-Plate, the bottom of the upper chambers of which was beforehand covered with Matrigel, modeling the extracellular matrix. An assessment of the cell index showed that TGF-β significantly reinforced invasion of tumor cells: the cell indexes of TGF-β-treated cells reached values of 1.5 and 3.3 at 24 h and 48 h, respectively, versus 1.2 and 2.5 in the control group (Figure 5D). The treatment of TGF-β-stimulated cells by SM significantly inhibited their invasion through Matrigel, leading to decrease of this parameter up to the level of control cells (Figure 5D). It should be emphasized, that SM itself also decreased the invasion capacity of A549 cells in the absence of EMT stimulus, declining the cell index by 1.6 times compared to the untreated control group at 24 h, however, this effect disappeared to the end of the experiment (48 h) (Figure 5D). Thus, SM effectively blocked the invasion of both rested and TGF-β-simulated A549 cells, which showed a high potency of SM as the probable inhibitor of tumor metastasis.

#### 2.3.3. SM Shifted the Expression of EMT-Associated Markers to Epithelial Ones

According to a recent guideline of the EMT International Association (TEMTIA), the EMT status of the cells has to be assessed on the basis of several characteristics due to complexity and plasticity of the EMT program, among which the evaluation of molecular markers should be considered as essential element of such analysis [1]. In order to implement this recommendation, we further analyzed the effect of SM on the expression of E-cadherin, vimentin and fibronectin in TGF-β-stimulated A549 cells.

E-cadherin is a well-known epithelial marker, being an essential component of adherent junctions, repression of expression of which causes the loss of epithelial-like cobblestone morphology of cells and its replacement by a spindle-shaped mesenchymal phenotype [5]. Vimentin and fibronectin, in turn, are considered as mesenchymal markers [5]. It was found that the overexpression of vimentin led to the replacement of cytokeratin-type intermediate filaments, being specific for epithelial-like cells, by vimentin type and this process always accompanied EMT and tightly associated with malignancy [44]. The latter marker, fibronectin, is an important extracellular matrix glycoprotein, playing the key role in cell adhesion and migration, the expression of which is up-regulated in mesenchymal-like cells [45].

Performed analysis showed that the incubation of A549 cells in the presence of TGF-β significantly suppressed expression of E-cadherin and up-regulated expression of vimentin and fibronectin by 2.0-, 7.9- and 4.2-fold compared to untreated control (Figure 6A), which indicates the triggering of EMT. The treatment of TGF-β-activated tumor cells by SM effectively shifted the expression of EMT-associated markers to epithelial-type one: it was shown that SM increased the expression of E-cadherin by 3.3-fold and decreased the expression of fibronectin by 1.9-fold compared to TGF-β-stimulated cells (Figure 6A). The expression of vimentin was also suppressed in TGF-β-treated cells in response to SM; however, this effect was statistically insignificant (Figure 6A).

Next, in order to double-check the revealed ability of SM to affect the expression of EMT-associated markers, we evaluated the effect of SM on the expression of zonula occludens 1 (ZO-1), a protein component of the tight junctions. Flow cytometry analysis showed that stimulation of A549 cells by TGF-β decreased expression of ZO-1, whereas treatment of these cells by SM effectively blocked mentioned effect: the intensity of fluorescence of the cells incubated with both TGF-β and SM was similar to that of the control untreated cells (Figure 6B). Interestingly, that SM itself increased expression of ZO-1 in A549 cells in the absence of EMT stimulus (Figure 6B), which agreed well with recently revealed ability of pentacyclic triterpenoids to up-regulate expression of this protein in other models [46,47].

Thus, obtained results clearly showed that SM effectively blocked TGF-β-induced changes in the expression of EMT-associated molecular markers in tumor cells and demonstrated the expediency of its further investigation as promising EMT inhibitor.

#### 2.3.4. Network Pharmacology Revealed JNK1 and MMP-2/-9 as the Probable Primary EMT-Associated Targets of SM

It is widely known that natural metabolites and their chemical derivatives, including CETs, are characterized by a multitarget mode of action [25], which determines the need for integrating system approaches in the search for molecular targets of such compound, e.g., omics technologies and network-based analysis. Due to lack of understanding of molecular mechanism of CETs’ action on EMT, the next task of the present study was to identify probable primary protein targets of SM, being associated with EMT program, by using network pharmacology approach. To address this, we predicted probable protein targets of SM by using two independent web servers SwissTargetPrediction and Polypharmacology Browser PPB2, which computed a structural similarity of SM with known ligands, using two different methods, such as a combination of 2D and 3D similarities [48] and a nearest neighbor search with Naïve Bayes machine learning approach based on 2D molecular fingerprint ECfp4 (the extended connectivity fingerprint, encoding a circular substructure of diameter four bonds) [49], respectively. Predicted molecular targets of SM are listed in Appendix A. Next, we questioned which proteins from the obtained list of probable SM’s targets were most associated with EMT. In order to understand this, we firstly identified EMT-related genes, being up- or down-regulated (fold change > 1.5, *p* < 0.05) in A549 cells, stimulated by TGF-β for 24 h, in comparison with unstimulated cells, by re-analysis of microarray dataset GSE17708, being deposited previously in the Gene Expression Omnibus by Sartor et al. [50]. After that, a gene association network was reconstructed based on the Search Tool for the Retrieval of Interacting Genes/Proteins (STRING) database, where revealed EMT-associated differentially expressed genes and predicted targets of SM were used as input data. As a result, a dense network, consisted from 366 proteins/genes and 1446 interconnections between them, was obtained (Figure 7A). Further ranking of predicted targets of SM by the level of their interconnection within the network revealed proteins that are the most associated with EMT-related regulome, top 10 of which is listed in Figure 7A.

In order to verify obtained data, an ability of SM to interact with the first three hit targets, including matrix metalloproteinases-9 and -2 (MMP-9/-2) and c-Jun N-terminal protein kinase 1 (JNK1; also known as MAPK8) was evaluated by using molecular docking approach. Our results showed that analyzed proteins can be considered as probable primary targets of SM: investigated triterpenoid can fit into their ligand binding pockets with relatively low binding energies (ΔG < 7.0 kcal/mol) (Figure 7B) and form similar interaction patterns in the structures of mentioned proteins with their known inhibitors (Figure 7C,D), including formation of hydrogen bonds with key amino acid residues needed for catalytic activity of these proteins. In the case of MMP-9, SM was found to form strong hydrogen bond with His230 (3.2 Å), imidazole ring of which coordinates catalytic zinc ion [51]; the top-scoring binding pose of SM in MMP-2 is characterized by the formation of hydrogen bond between the keto group in ring A and catalytic zinc ion (Figure 7D), whereas, in the case of JNK1, SM forms hydrogen bond with Lys55, which is involved in ATP binding and maintaining kinase architecture of JNK1 [52] (Figure 7D). Moreover, obtained complexes SM-MMP-9 and SM-JNK1 were stabilized by additional two hydrogen bonds with Gln227 (2.61 Å, 3.17 Å) and one hydrogen bond with Asn114 (3.16 Å), respectively (Figure 7D).

Thus, performed network pharmacology analysis revealed that SM can control EMT of tumor cells hypothetically by its direct interactions with MMP-2/-9 and JNK1, playing an important role in EMT program. In order to elucidate this mechanism more precisely, a detailed biochemical or biophysical analysis is further required.

#### 2.3.5. SM Effectively Inhibited a Lung Metastasis of Melanoma B16 Cells In Vivo

Given the revealed ability of SM to effectively block EMT of tumor cells in vitro and tight association of EMT with metastasis of malignant cells [1], we further questioned whether SM inhibits metastatic growth of B16 melanoma in murine model. Since SM suppresses reinforcement of metastatic potential of tumor cells, associated with TGF-β-driven EMT, we first asked whether aggressiveness of B16 melanoma cells depends on TGF-β production and signaling. To address this, we compared whole genome expression profile of B16 cells with that of D5.1G4 murine melanoma cells, which are characterized by poor tumorigenic and metastatic capacities, and identified 827 differentially expressed genes (DEGs) (|Log_2_(Fold Change)|>2, *p* < 0.05). Revealed DEGs were further used as input data to reconstruct gene regulatory network associated with highly aggressive phenotype of melanoma cells (Appendix A), followed by the analysis of the network with subsequent identification of key nodes being the most interconnected with neighbors in the regulome (Figure 8A, left panel). Our analysis revealed that TGF-β (encoded by *Tgfb1* gene) can be considered as master regulator of aggressive growth and metastasis of B16 melanoma cells: it was found that *Tgfb1* was involved in top 10 key nodes, displaying 23 interconnections with partner genes in the network, including other identified key nodes Ccl2, Cd34, Ccl5, Pdgfrb and Col1a2 (Figure 8A, left panel). Moreover, it was demonstrated that Tgfb1 gene expression was 16.8-fold up-regulated in B16 cells compared to its poorly tumorigenic D5.1G4 counterpart (Figure 8B, right panel). Revealed key regulatory role of TGF-β in maintaining of high aggressiveness of B16 melanoma cells agreed well with published data. Previously, it was shown that these cells secrete high amount of TGF-β [53,54], which can trigger EMT of melanoma cells in tumor tissue [54]. Furthermore, the blockage of TGF-β by selective peptide inhibitors or its silencing by shRNA or siRNA in B16 cells significantly inhibited tumor growth, metastasis and the sensitivity of melanoma to immunotherapy in vivo [55,56,57]. Thus, TGF-β was found to play an important role in B16 melanoma progression and, therefore, the selection of this metastatic model of B16 melanoma was suitable for further evaluation of the antimetastatic and anti-EMT activities of SM in vivo.

Mice bearing B16 cells were administered intraperitoneally with SM at dosage of 25 mg/kg twice a week starting from the fourth day after tumor transplantation; totally six injections were carried out (Figure 8B). Mice were sacrificed on day 21 of tumor growth followed by the evaluation of the number of surface metastases in lungs and the histological analysis of the lung tissues.

It was found that the administration of SM led to 2.4- and 2.6-fold decrease in the number of surface metastases in the lungs of B16 melanoma-bearing mice compared to untreated and vehicle-treated animals, respectively (Figure 8C). Histological examination showed that metastatic foci of B16 melanoma in the lungs of control and experimental mice were rounded with clear boundaries and represented by the polymorphic or spindle shaped atypical cells containing the brown pigment melanin (Figure 8D) (for more detailed images, please see primary data in Appendix A). Further calculation of the area of internal metastasis of B16 melanoma in the lungs revealed that the administration of SM reduced this parameter by 2.5-times compared to the untreated control, however, the differences among these groups were statistically insignificant (Figure 8E). Interestingly, in spite of the absence of inhibitory effects on the dissemination of melanoma B16 cells, the vehicle was found to reduce the area of internal metastases in the lungs by 2.2 times compared to the control (statistically insignificant). We suppose that this effect of vehicle can be explained by the presence of moderate anti-tumor activity of Tween-80, revealed previously by other research groups [58]. Thus, obtained results showed that SM effectively prevented the development of new metastatic foci hypothetically by the reducing the potential of tumor cells to colonize a new organ; however, the used dosage of the triterpenoid (25 mg/kg) was insufficient for effective blockage of growth of already formed metastases.

Next, in order to estimate the effect of SM on the expression of markers associated with EMT and tumor cell invasion in the metastatic foci, fluorescence-based immunohistochemistry for E-cadherin and MMP-9 was performed. As shown in Figure 8F, the administration of SM to B16 melanoma-bearing mice significantly increased expression of E-cadherin in lung metastases: the fluorescent intensity in this group was found to increase by 3.9-fold compared to untreated control group. Furthermore, a 4.3-fold decrease of fluorescence intensity was observed in the sample, obtained from the lungs of SM-treated animals, stained by anti-MMP-9 antibodies, in comparison with the control (Figure 8F). The administration of vehicle alone to mice with B16 melanoma was found to decrease the expression of E-cadherin and MMP-9; however, the effect of vehicle on the latter parameter was significantly lower than that of SM (Figure 8E) (raw microscopy images of mentioned samples can be found in Appendix A). Thus, performed immunohistochemistry analysis clearly confirmed the ability of SM to modulate expression of EMT-associated markers not only in tumor cells, but also in metastases in animal model.

## 3. Discussion

Lung cancer is the most widespread and deadliest cancer worldwide, accounting for more than 14 million new cases and more than 1.6 million deaths annually [59]. In Russia, respiratory organs cancer is the most common type of oncologic diseases—according to Avxentyeva et al., in total, 185,631 patients were diagnosed with trachea, bronchus and lung cancer in 2016 in Russia with 51,769 incident cases and 51,476 deaths [60]. The one of the important cause of such high mortality of patients, along with the late diagnosis, is the lack of drugs, being able to effectively block signals from tumor microenvironment, which stimulate motility, invasion and dissemination of tumor cells from the primary site to distant organs due to induction of epithelial-mesenchymal transition (EMT), leading to an acquisition of mesenchymal-like highly invasive phenotype by malignant cells [1,2,5]. The tight association of EMT with an increase of malignancy of lung adenocarcinoma is evidenced by the fact that overexpression of mesenchymal-type related N-cadherin and Twist1 was correlated with a shorter overall survival [61] as well as vimentin expression can serve as a prognostic biomarker for the metastases [62] in non-small cell lung adenocarcinoma patients. Given this interconnection, the inhibition of EMT can be considered as a promising strategy to reinforce the efficacy of conventional antitumor therapy.

For this reason, extensive efforts have been devoted to the search and development of low molecular weight inhibitors of EMT, among which pentacyclic triterpenoids are considered as a candidate platform [10]. To date, the ability to suppress EMT of different tumor cells has been shown for a wide range of natural triterpenoids, including oleanolic [12,14], asiatic [13,18,19], ursolic [16,21,22,63], betulinic [17] acids and celastrol [15] and some triterpenoid saponins [23,24]. However, despite of this, the effects of semisynthetic triterpenoids on EMT process is poorly investigated.

In this work, we concentrated on the question whether cyano enone-containing semisynthetic triterpenoids or steroids can modulate EMT of human lung adenocarcinoma A549 cells, stimulated by TGF-β. Performed screening of chemical derivatives of 18βH-glycyrrhetinic (SM, TM) and deoxycholic (pi-153, pi-156) acids, marked bioactivity of which was identified previously [35,36,37,41], for inhibition of motility of A549 and B16 cells revealed SM as a hit compound, significantly suppressing in non-toxic concentration the migration capacity of both cell lines (Figure 3). The remaining compounds either showed a lower level of bioactivity compared to SM (pi-153) or were unable to affect motility of tumor cells at all (TM, pi-156). The obtained results are in line with published data: recently, we found that SM at identical concentration (0.5 µM) effectively inhibited migration of RAW264.7 macrophages [64]; furthermore, SM’s structural analog CDDO-Me and its derivatives significantly reduced motility of both malignant [65,66,67,68,69] and non-malignant [70,71,72] cells at submicromolar concentrations (0.05–1 µM). Revealed differences in the level of inhibitory activity of investigated compounds against tumor cell motility are consistent with the level of their cytotoxicity. It was found that SM, the hit compound, showed the highest cytotoxic capacity in both used cell lines compared to other evaluated derivatives (Figure 2). In the case of steroids, pi-153, which inhibited motility of only A549 cells, exhibited a more pronounced cytotoxicity with respect to a panel of tumor cells [41] in comparison with pi-156, being unable to affect cell migration (IC_50_ = 18.3–31.1 µM versus 51.5–100 µM, respectively) [41]. The revealed relationship, along with the absence of toxic effects of the compounds in the concentrations used for the scratch assay (Figure 2, Figure 3A,B and Figure 4A), shows the promising antitumor potential of SM and pi-153, which can inhibit not only tumor cell viability, but also their motility.

Further investigation demonstrated that SM effectively blocked EMT of A549 cells, stimulated by TGF-β: namely, the acquisition of fibroblast-like morphology by the cells (Figure 5A), their high migration and invasion capabilities (Figure 5B–D) and expression of mesenchymal marker fibronectin (Figure 6A), whereas expression of epithelial markers E-cadherin and ZO-1 was significantly up-regulated in TGF-β-stimulated A549 cells treated with SM (Figure 6).

It is known that overexpressed fibronectin can interact with several integrin receptors and syndecans exposed on the surface of tumor cells and, thereby, activate various signaling cascades, which promote tumor cell proliferation, invasion, metastasis and therapy resistance [73]. In previous works, it was shown that silencing of fibronectin by siRNA or shRNA led to significant suppression of motility and invasion of A549 lung adenocarcinoma [74] and 1205Lu melanoma [75] cells and growth of SiHa cervical carcinoma xenografts in nude mice [76]. Given the revealed ability of SM to down-regulate expression of fibronectin (Figure 6A) along with up-regulation of E-cadherin (Figure 6A) and ZO-1 (Figure 6B), being the essential components of adherent and tight junctions, respectively, as well as marked inhibition by SM of clonogenic activity of A549 cells (Figure 3C), we anticipated that SM can affect metastasis development in vivo. Indeed, performed experiment on metastatic model of murine B16 melanoma clearly showed that SM inhibited dissemination of melanoma cells in vivo: the number of surface metastasis in lungs of B16 melanoma-bearing mice treated by SM was significantly lower than that of vehicle-treated control (Figure 8B). Furthermore, injections of SM were found to decrease the area of internal metastasis in lungs, however, these changes were statistically insignificant compared to untreated and vehicle-treated groups (Figure 8D). We suppose that observed insufficient ability of SM to suppress growth of metastatic foci in this model can be explained by the high aggressiveness of B16 melanoma [77] and probably by unoptimized pharmacological composition (10% Tween-80). It was demonstrated that the vehicle has own moderate anti-tumor potential (Figure 8D) that agreed well with [58]; beside this, we also assume probable inability of the vehicle to maintain the concentration of SM in blood plasma needed for marked therapeutic effects. In spite of this, it was found that injections of SM markedly increased expression of E-cadherin in metastatic foci (Figure 8E), which is consistent with the revealed ability of this compound to decrease the number of surface metastasis (Figure 8B). It is known that E-cadherin mediates intercellular contacts and, therefore, impedes metastasis [78]. Moreover, fluorescence-based immunohistochemistry analysis also showed that metastatic foci in lungs of SM-treated mice are characterized by a low expression of MMP-9, which can degrade extracellular matrix and, therefore, plays a crucial role in tumor cell invasion [79] (Figure 8E). Thus, our findings clearly demonstrate the high anti-metastatic potential of SM, however, in order to elucidate this activity more precisely further study of its pharmacokinetics and optimization of its pharmaceutical formulation are needed. Revealed ability of SM to inhibit metastasis of B16 melanoma in murine model is in line with published data. Previously, it was shown that other CETs effectively inhibited metastasis of murine B16 melanoma, 4T1 mammary carcinoma and prostate transgenic adenocarcinoma in liver, lungs and pelvic lymph nodes, respectively [80,81,82].

An important question that we also address in this study was which mechanism can underlie the observed inhibitory activity of SM on TGF-β-driven EMT of tumor cells. According to published data, CETs either did not affect TGF-β/Smad signaling [83] or vice versa activated it both in tumor [84,85] and non-malignant [86] cells. Considering this fact, we speculate that SM inhibits TGF-β-stimulated EMT of A549 cells TGF-β/Smad-independently. In order to shed some light on this issue, we used network pharmacology approach, which revealed MAP kinase JNK1 and matrix metalloproteinases MMP-2 and MMP-9 as probable primary targets of SM, associated with its anti-EMT activity (Figure 7). It is known that TGF-β can induce EMT not only by TGF-β/Smad signaling axis, but also by non-Smad pathway: activated TGF-β receptor complex can stimulate polyubiquitination of tumor necrosis receptor-associated factors TRAF4/6 followed by the recruitment of TGF-β-activated kinase TAK1 and subsequent activation of JNK1 signaling [42], which plays an important role in the regulation of EMT [87]. Matrix metalloproteinases, being key EMT-related genes, can be also involved in EMT induction. Previously, it was found that stimulation of cervical carcinoma A431 cells by recombinant MMP-9 led to an acquisition of spindle-like phenotype by the cells; furthermore, knockdown of MMP-9 by siRNA significantly reduced expression of mesenchymal markers vimentin and fibronectin in these cells [88].

Revealed probable ability of SM to bind to active sites of JNK1 and MMP-2/-9 are quite consistent with previous reports. It was found that pentacyclic triterpenoids can inhibit JNK1 activity: oleanolic acid acetate, isoxazole ursolic amide (IUA) and celastrol were shown to significantly inhibit phosphorylation of c-Jun, a downstream effector of JNK1, in leukemia THP-1, osteosarcoma HOS and Saos-2 cells and cerebral ischemia, respectively [89,90,91]. Moreover, molecular simulations performed by Negi et al. demonstrated that ursolic acid can snugly fit into the active site of JNK1 [92]. In the case of matrix metalloproteinases-2/-9, primary effects of triterpenoids on their activity, as far as we know, have not been yet published, however, recently, Preciado et al. reported that betulinic, oleanolic and ursolic acids can bind to catalytic site of snake venom metalloproteinases, being similar to those in human matrix metalloproteinases, and significantly inhibit their enzymatic and biological activities [93,94]. Furthermore, CDDO-Me and its analog RTA408 were found to decrease expression of MMP-9 in various models, including tumor necrosis factor α- (TNF-α-), receptor activator of nuclear factor kappa-B ligand- (RANKL-) and interleukin 1β- (IL-1β)-stimulated chronic myelogenous leukemia KBM-5 cells, RAW264.7 macrophages and rat brain astrocytes [72,95,96], respectively, as well as primary mammary tumor cells, isolated from polyoma middle T mice [97], which is consistent with revealed ability of SM to inhibit expression of MMP-9 in metastatic foci in lungs of B16 melanoma-bearing mice (Figure 8E). It should be emphasized, however, that prediction of JNK1 and MMP-2/-9 as EMT-associated protein targets of SM was carried out using computer modeling approach and, therefore, these findings only hypothesize about probable molecular mechanism of inhibitory effect of SM on EMT. Further verification of direct interactions of SM with mentioned proteins by using conventional biochemical or biophysical methods are needed.

## 4. Material and Methods

### 4.1. Chemicals and Reagents

The chemical synthesis of investigated derivatives of 18βH-glycyrrhetinic (SM, TM) and deoxycholic (pi-153, pi-156) acids and their characterization by chemical analysis and nuclear magnetic resonance have been described in our earlier works [35,36,41]. The compounds were dissolved in DMSO at 10 mM, and stock solutions were kept at −20 °C until further use. Recombinant TGF-β (CYT-716) was purchased from ProSpec-Tany TechnoGene Ltd. (Ness-Ziona, Israel). Primary antibodies used in this study were purchased from Abcam (Cambridge, MA, USA), including anti-E-cadherin [M168] (ab76055), anti-MMP-9 (ab38898) and anti-ZO-1 (ab216880). Secondary antibodies Alexa Fluor^®^ 488-conjugated IgG (ab150077) and DyLight™ 488-conjugated IgG (2632020) were obtained from Abcam (Cambridge, MA, USA) and Sony Biotechnology Inc. (San Jose, CA, USA), respectively.

### 4.2. Cell Cultures

The human A549 lung adenocarcinoma and murine B16 melanoma cells were obtained from the Russian Cell Culture Collection (St. Petersburg, Russia). The human non-malignant gingival fibroblasts (HGF) were kindly provided by Dr. Pavel P. Laktionov (Institute of Chemical Biology and Fundamental Medicine, Siberian Branch of the Russian Academy of Sciences (SB RAS), Novosibirsk, Russia). The cells were cultured in Dulbecco’s modified Eagle’s medium (DMEM) (Sigma-Aldrich Inc., St. Louis, MO, USA) supplemented with 10% (*v*/*v*) heat-inactivated fetal bovine serum (FBS) (HyClone^TM^, GE Healthcare Life Sciences, South Logan, UT, USA), 100 units/mL penicillin, 100 µg/mL streptomycin and 0.25 µg/mL amphotericin. The cells were incubated at 37 °C in a humidified air atmosphere, containing 5% CO_2_ (hereafter referred to as standard conditions).

### 4.3. Mice

Female C57Bl/6 mice (20–24 g) were obtained from the Vivarium of Institute of Chemical Biology and Fundamental Medicine SB RAS (Novosibirsk, Russia). Mice were housed in plastic cages (10 animals per cage) under normal daylight conditions. Water and food were provided ad libitum. Experiments were carried out in accordance with the European Communities Council Directive 86/609/CEE. The experimental protocols were approved by the Committee on the Ethics of Animal Experiments at the Institute of Cytology and Genetics SB RAS (Novosibirsk, Russia) (protocol No. 51 from 23 May 2019).

### 4.4. Cell Viability Analysis

A549, B16 and HGF cells were seeded in 96-well plates at densities of 10^4^, 2 × 10^4^ and 8 × 10^3^ cells/well, respectively, and incubated under standard conditions for 24 h. After that, the medium was replaced with serum-free DMEM containing investigated derivatives of 18βH-glycyrrhetinic and deoxycholic acids at 0.06–1 µM and 10–100 µM, respectively, (tumor cells), SM at 0.1–10 µM (HGF) or corresponding concentrations of DMSO (control) followed by the incubation for subsequent 48 h (tumor cells) or 24 h (HGF) under standard conditions. Then, 3-(4, 5-dimethylthiazol-2-yl)-2, 5-diphenyltetrazolium bromide (MTT) (Sigma-Aldrich Inc., St. Louis, MO, USA) was added to each well at 0.5 mg/mL and the cells were incubated for additional 2 h. After that, the medium, containing MTT, was discarded from the wells and 100 µl of DMSO was added to each well to dissolve the formazan crystals followed by the measurement of the absorbance at a test and reference wavelengths of 570 nm and 620 nm, respectively, on a Multiscan RC plate reader (Thermo LabSystems, Helsinki, Finland).

### 4.5. Measurement of Cell Motility Using Scratch Assay

A549 and B16 cells were seeded in triplicate in 24-well plates at a density of 1.3 × 10^5^ cells/well in antibiotic-free DMEM supplemented with 10% FBS and incubated under standard conditions for 24 h. Thereafter, vertical scratches were made on the cell monolayers with 10 µl pipette tips followed by two washing steps with phosphate buffered saline (PBS) to remove floating cells and cell debris and the addition of investigated compounds (0.5 µM and 50 µM for semisynthetic triterpenoids and steroids, respectively) and/or TGF-β (50 ng/mL) in antibiotic- and serum-free DMEM. At time points 0 h, 24 h and 48 h the scratched cell monolayers were visualized using ZEISS Primo Vert invert microscope equipped with ZEISS AxioCam ERc5s camera (both instruments from Carl Zeiss Microscopy GmbH, Jena, Germany). The wound closure was calculated by normalization of the area of scratch occupied by the cells in experimental groups at each time points to this parameter in control group using ImageJ software (National Institutes of Health, Bethesda, MD, USA). Totally, 4–5 regions on a single scratch were analyzed.

### 4.6. Measurement of Cell Motility and Invasion Using xCELLigence Platform

A549 cells were seeded in tetraplicate at a density of 2 × 10^4^ cells/well in the upper chamber of CIM-Plate either uncovered (migration assay) or covered with Matrigel (10 µL per well, diluted 1:40 with cold serum-free DMEM) (invasion assay) in serum-free DMEM in the presence or absence of SM (0.5 µM) and TGF-β (50 ng/mL). In order to stimulate motility of the cells, DMEM supplemented with 10% FBS as chemoattractant was placed to the lower chamber of CIM-Plate. Thereafter, the electrical impedance (cell index) of sensor electrodes mounted at the lower side of porous membrane separating the upper and lower chambers of the plate was further measured using xCELLigence RTCA DP instrument (ACEA Biosciences, San Diego, CA, USA) every 1 h for 48 h.

### 4.7. Colony Forming Assay

A549 cells were seeded in 96-well plates in 5 technical replicates at a density of 200 cells per well and treated with SM at concentrations of 0.125–0.5 µM for 14 days under standard conditions. At the end of the treatment, developed cell colonies were fixed with 4% paraformaldehyde followed by the staining of the colonies with crystal violet (0.1% *w*/*v*) and visualization using ZEISS Primo Vert invert microscope. The number of cell colonies in the wells was calculated using ImageJ tool.

### 4.8. Analysis of Cell Morphology

A549 cells were seeded in 96-well plates at a density of 7 × 10^3^ cells per well in tetraplicate in 100 µL of DMEM supplemented with 10% FBS followed by the incubation for 24 h under standard conditions. Next, the medium was replaced by fresh serum-free DMEM, containing needed concentrations of TGF-β and/or SM (0.5 µM), and the cells were incubated under standard condition for further 48 h. Thereafter, cell morphology was analyzed using ZEISS Primo Vert invert microscope. The quantitative assessment of changes in morphology of the cells was carried out by comparison of spindle-like cell morphology (SLCM) score in experimental and control groups, calculated according to the formula:(1)$$\mathrm{SLCM}\;\mathrm{score}=\frac{L_{long\;axis}}{L_{short\;axis}},$$
where *L* is the length of cells, long and short axes are longitudinal and transverse axes of the cells. Totally, the measurement of 200 random cells per group was performed using ImageJ software.

### 4.9. Analysis of Gene Expression by Reverse Transcription PCR (RT-PCR)

A549 cells were seeded in 24 well plates at a density of 6 × 10^4^ cells per well in tetraplicate in DMEM supplemented with 10%FBS followed by the incubation for 24 h under standard conditions. Thereafter, the cells were treated by SM at 0.5 µM for 48 h followed by the extraction of total RNA in Trizol (Ambion, Austin, TX, USA) according to the manufacturer’s protocol. The first-strand of cDNA was synthesized by using 1.5 µg of total RNA, 2 µM of random hexa primers and M-MuLV-RH reverse transcription kit (Biolabmix, Novosibirsk, Russia). The PCR was performed in tetraplicate using a BioMaster HS-qPCR SYBR Blue (2×) (Biolabmix, Novosibirsk, Russia) in a final volume of 20 µL, containing 5 µL of cDNA template and 0.4 µM of gene specific primers (Appendix A) used Real-time CFX96 Touch (Bio-Rad Laboratories Inc., Hercules, CA, USA). The PCR conditions were as follows: 95 °C, 5 min; 95 °C, 30 s; 60 °C, 10 s; 72 °C, 20 s. The last three steps were repeated 40 times. The housekeeping gene *GAPDH* was used as a reference gene. PCR specificity was controlled using melting curve analysis. Calculation of relative gene expression normalized to glyceraldehyde-3-phosphate dehydrogenase (GAPDH) was carried out according to the comparative threshold cycle (ΔΔC_T)_ method.

### 4.10. Flow Cytometry

A549 cells were seeded in 6 well plates at a density of 5 × 10^5^ cells/well in DMEM supplemented with 10% FBS and incubated for 24 h under standard conditions. Next, the medium was replaced by fresh serum-free DMEM containing TGF-β (50 ng/mL) and/or SM (0.5 µM) and the cells were incubated for further 48 h. At the end of incubation time, the cells were collected and fixed by the Fixation Medium A of Fix&Perm^®^ Cell Permeabilization Reagent (Invitrogen, Frederick, MD, USA) followed by the washing of cells by PBS and their incubation for 1 h with the specific anti-ZO-1 monoclonal antibodies (1:100) dissolved in Permeabilization Medium B of Fix&Perm^®^ Cell Permeabilization Reagent. After that, the cells were washed by PBS and stained by DyLight™ 488-conjugated secondary antibodies for subsequent 30 min followed by the next washing by PBS and analysis by flow cytometry using an ACEA NovoCyte^TM^ flow cytometer (ACEA Biosciences Inc, San Diego, CA, USA). For each sample 10,000 events were acquired.

### 4.11. Network Pharmacology

On the first step of the study the probable primary targets of SM were predicted by Polypharmacology Browser PPB2 [98], using Naïve-Bayes machine learning approach with nearest neighbors search based on the extended connectivity fingerprint of ECfp4, and SwissTargetPrediction [99], according to the developer’s instructions [49,100]. Top 20 and top 15 predicted targets of SM from PPB2 and SwissTargetPrediction servers, respectively, were further used for subsequent analysis. Next, in order to reveal the regulome, which controls TGF-β-induced EMT of A549 cells, the gene expression profile of GSE17708 was uploaded from the Gene Expression Omnibus [101] and the differentially expressed genes (DEGs) between TGF-β-stimulated and unstimulated cells (fold change > 2; *p* < 0.05) were computed using GEO2R tool [102,103]. Finally, the gene association network based on revealed probable targets of SM and EMT-associated DEGs were reconstructed using the STRING database [104] with a confidence score >0.7 and visualized by Cytoscape 3.6.1. Only nodes with ≥1 interconnections within the network were considered. The ranking of predicted SM’s targets according to their level of interconnection into the network was further carried out using the NetworkAnalyzer plugin [105].

### 4.12. Molecular Docking

The docking of SM with JNK1 (Protein Data Bank (PDB) ID: 4AWI), MMP-2 (PDB ID: 1HOV) and MMP-9 (PDB ID: 1GKC) were performed using Autodock Vina [106]. The crystal and solution structures of the proteins (JNK1/MMP-9 and MMP-2, respectively) were uploaded from the Research Collaboratory for Structural Bioinformatics (RCSB) Protein Data Bank (https://www.rcsb.org/). The co-crystalized ligands and water molecules were extracted from the PDB files of the proteins followed by the addition of polar hydrogen and Gasteiger charges into the protein structures using AutoDock Tools 1.5.7. The two-dimensional structure of SM was converted into 3D form and its geometry was optimized with the MMFF94 force field using Marvin Sketch 5.12 and Avogadro 1.2.0, respectively. All the rotatable bonds within SM were allowed to rotate freely. The used docking parameters are listed in Table 1 (PDB IDs are identifiers of three-dimensional structures of proteins in PDB database).

The best molecular interactions, characterized by the presence of hydrogen bonds with proteins’ key residues needed for catalytic activity and the formation of interaction network similar to that of known inhibitors, were identified. The revealed hit docked complexes were further visualized by BIOVIA Discovery Studio Visualizer 17.2.0 (Dassault Systemes, Cedex, France) and LigPlot+ 1.4.5 (European Bioinformatics Institute, Cambridge, UK) in 3D and 2D forms, respectively.

### 4.13. Bioinformatic Analysis of Melanoma Regulome, Associated with Highly Aggressive Phenotype

To decipher probable master regulators of melanoma progression, the whole genome expression profiles of highly aggressive B16 and poorly tumorigenic D5.1G4 murine melanoma cells were compared by re-analysis of GSE69908 cDNA microarray dataset, deposited previously in Gene Expression Omnibus database by K. M. Hargadon [107], using GEO2R tool. Further, the DEGs between B16 and D5.1G4 cells (|Log_2_(Fold Change)|>2, *p* < 0.05) were computed followed by reconstruction of gene regulatory network from revealed DEGs using STRING database (confidence score > 0.4). Finally, analysis of gene regulatory network was performed using the NetworkAnalyzer plugin and top 10 key nodes, being the most interconnected with partner genes in the regulome, were identified. The level of interconnection of revealed key nodes with gene regulatory network and between each other as well as their level of expression were visualized by Circos [108,109] and Morpheus platform [110], respectively.

### 4.14. Tumor Transplantation and Design of Animal Experiment

Metastatic model of tumor progression was induced by intravenous (i.v.) injection of B16 melanoma cells (10^6^ cells/mL) suspended in 0.2 mL of saline buffer into the lateral tail vein of C57Bl/6 mice. On day 4 after tumor transplantation, mice were assigned into three groups (*n* = 10 per group): (1) mice without treatment (control); (2) mice received intraperitoneal (i.p.) injections of 10% Tween-80 (vehicle); and (3) mice received i.p. injections of SM in 10% Tween-80 at a dose of 25 mg/kg. The treatment was carried out twice a week. The total number of injections was six. Mice were sacrificed on day 21 after tumor transplantation and the lungs were collected for calculation of surface and internal metastases.

### 4.15. Analysis of Number of Surface Metastases, Histology and Fluorescence-Based Immunohistochemistry

Surface metastases in the lungs of B16 melanoma-bearing mice were counted using a binocular microscope. For the histological study, the lung specimens were fixed in 10% neutral-buffered formalin (BioVitrum, Moscow, Russia), dehydrated in ascending ethanols and xylols and embedded in HISTOMIX paraffin (BioVitrum, Moscow, Russia). The paraffin sections (5 μm) were sliced on a Microm HM 355S microtome (Thermo Fisher Scientific, Waltham, MA, USA) and stained with hematoxylin and eosin. The percentages of the internal metastases areas were determined relative to the total areas of lung sections using Adobe Photoshop software (Adobe Systems Inc., San Jose, CA, USA). Ten random fields from lung specimens of ten mice in each group (totally, 100 testing fields) were studied.

For the immunohistochemical study, the colon sections (3–4 μm) were deparaffinized and rehydrated. Antigen retrieval was carried out after exposure in a microwave oven at 700 W. The samples were incubated with the anti-E-cadherin or anti-MMP-9 specific primary antibodies according to the manufacturer’s protocol. Then, the sections were incubated with secondary Alexa Fluor^®^ 488-conjugated antibodies and embedded in Fluoromount-GTM Mounting Medium (Invitrogen, Thermo Fisher Scientific, Waltham, MA, USA). All the images were examined and scanned using Axiostar Plus microscope equipped with fluorescent lamp HBO 50W/AC L1 (Osram, Munich, Germany) and Axiocam MRc5 digital camera (Zeiss, Oberkochen, Germany) at magnifications of ×100 (hematoxylin and eosin images) and ×200 (fluorescence-based immunohistochemistry). Images were processed using ZEN SP2 (Zeiss, Oberkochen, Germany) and ImageJ software. Fluorescence intensity score was calculated according to the formula:(2)$$Fluorescence\;intensity\;score\;=\;\frac{I_{F}}{S},$$
where *I_F_* and *S* are fluorescence intensity and an area of analyzed metastatic foci, respectively, and visualized by Morpheus platform [110].

### 4.16. Statistical Analysis

All experiments were reproduced in duplicate or triplicate with 3–5 technical replicates. The statistical analysis was performed using the two-tailed unpaired t-test computed by Microsoft Excel (Microsoft Corp., Redmond, WA, USA). *p*-values < 0.05 were considered as statistically significant.

## 5. Conclusions

Performed screening of cyano enone-bearing derivatives of 18βH-glycyrrhetinic and deoxycholic acids revealed SM as a hit compound, able to significantly inhibit at non-toxic concentration motility of human lung adenocarcinoma A549 and murine melanoma B16 cells. Further analysis demonstrated that SM effectively blocked TGF-β-induced EMT of A549 cells by suppression of TGF-β-stimulated changes in cell morphology, inhibition of high migration and invasion capacities of TGF-β-treated A549 cells and switching the expression of EMT-associated markers to epithelial ones. Moreover, injections of SM were found to significantly decreased number of surface lung metastasis of B16 melanoma-bearing mice and lead to up-regulation of E-cadherin and down-regulation of MMP-9 in metastatic foci. Performed in silico analysis let us hypothesized that SM can control EMT of tumor cells by direct interactions with JNK1 and MMP-2/-9. Altogether, our results showed for the first time that cyano enone-containing triterpenoids can effectively inhibit TGF-β-driven EMT of tumor cells and revealed SM as a promising anti-metastatic candidate.

## Figures and Tables

**Figure 1 molecules-25-05925-f001:**
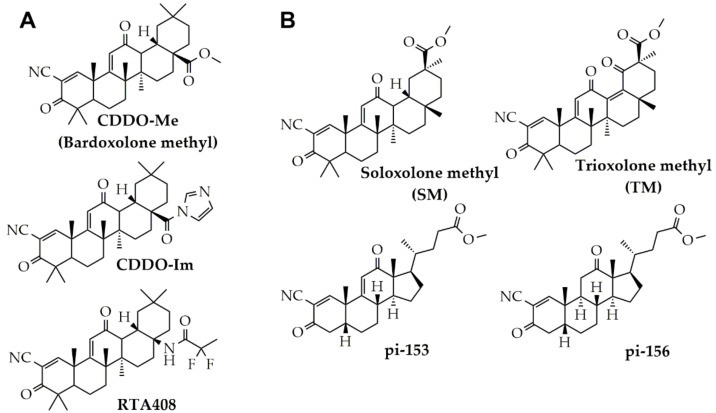
The chemical structures of cyano enone-containing semisynthetic triterpenoids and steroids. (**A**) Bioactive cyano enone-bearing triterpenoids. (**B**) Compounds under the study.

**Figure 2 molecules-25-05925-f002:**
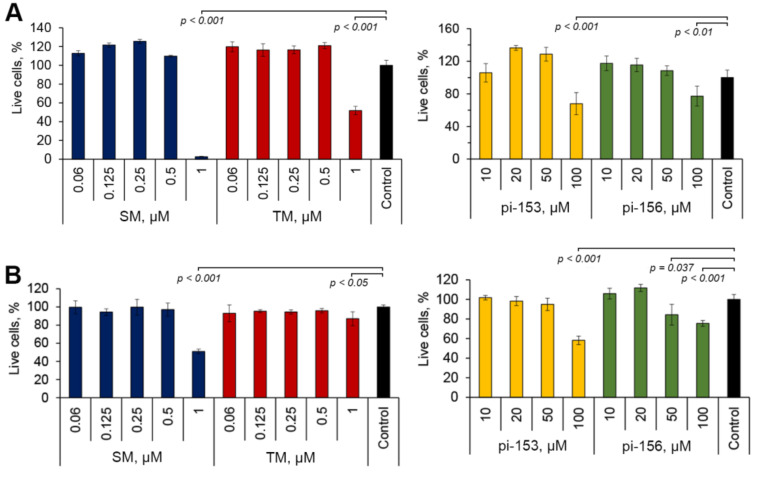
Cytotoxicity of cyano enone-containing semisynthetic triterpenoids (SM, TM) and steroids (pi-153, pi-156) with respect to murine melanoma B16 (**A**) and human lung adenocarcinoma A549 (**B**) cells. The cells were treated with mentioned compounds for 48 h followed by the measurement of cell viability by 3-(4, 5-dimethylthiazol-2-yl)-2, 5-diphenyltetrazolium bromide (MTT) assay. Error bars represent the standard deviation of three independent experiments performed in tetraplicate.

**Figure 3 molecules-25-05925-f003:**
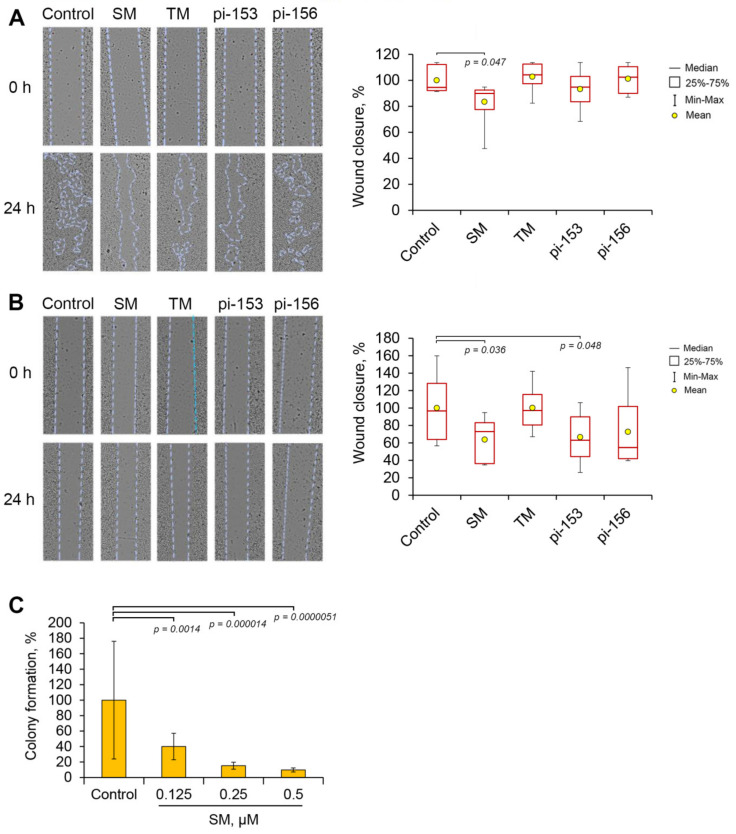
Effect of cyano enone-bearing semisynthetic derivatives of 18βH-glycyrrhetinic and deoxycholic acids on the motility of tumor cells. (**A**) and (**B**). Wound healing assay. The monolayers of B16 (**A**) and A549 (**B**) cells with scratches were treated with investigated compounds at non-toxic concentrations (SM and TM at 0.5 µM; pi-153 and pi-156 at 50 µM) for 24 h. Wound closure rates were calculated using ImageJ software. (**C**) Effect of SM on clonogenic activity of A549 cells. The cells were seeded at a low density and incubated with the presence of SM for 14 days followed by cell fixation and staining with crystal violet. The calculation of colony formation rate was carried out using ImageJ software.

**Figure 4 molecules-25-05925-f004:**
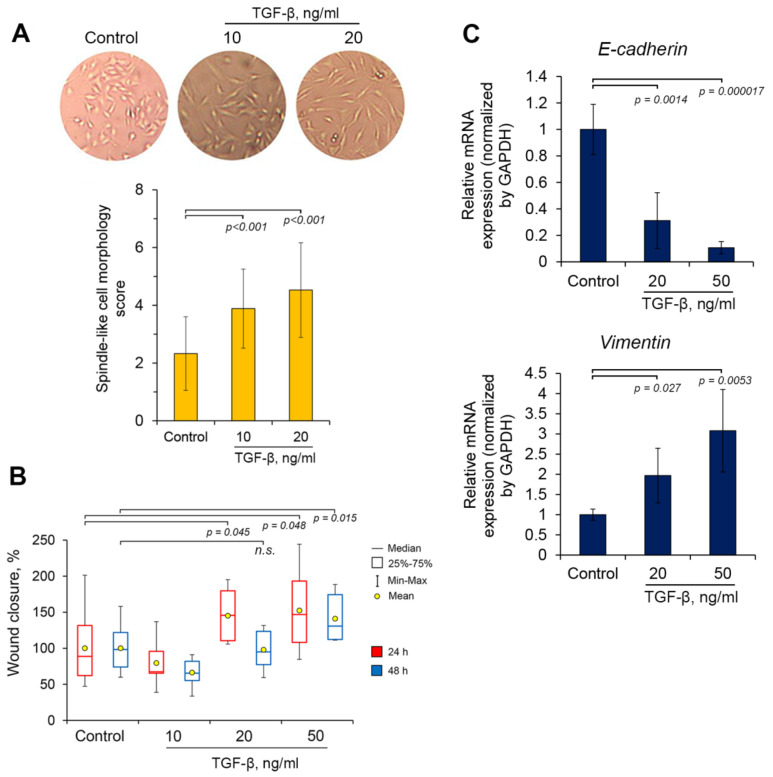
Selection of working concentration of transforming growth factor β (TGF-β) for induction of EMT of tumor cells. (**A**) TGF-β led to an acquisition of spindle-like phenotype by tumor cells. A549 cells were treated with TGF-β for 48 h followed by the evaluation of cellular morphology using phase contrast microscopy and the measurement of long and short axes of 200 random cells using ImageJ tool. (**B**) TGF-β increased motility of tumor cells. Scratched monolayers of A549 cells were treated with TGF-β for 24 h and 48 h after that wound closure rate was calculated using ImageJ software. (**C**) TGF-β switched expression of EMT-associated molecular markers to mesenchymal ones. Results of RT-PCR were from two independent experiments performed in tetraplicate. Relative expression levels were shown as mean ± standard deviation. Glyceraldehyde-3-phosphate dehydrogenase (GAPDH) was used as a housekeeping gene for normalization.

**Figure 5 molecules-25-05925-f005:**
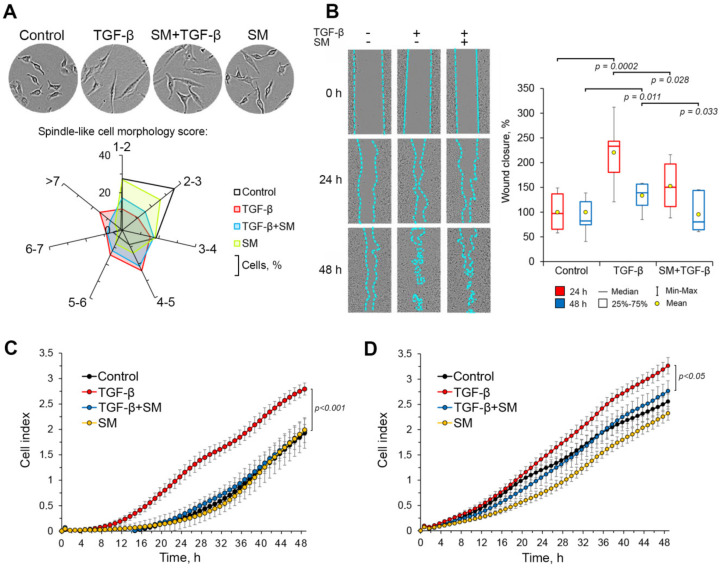
SM effectively inhibits TGF-β-stimulated EMT-associated processes in A549 cells. (**A**) SM increases the percentage of roundish-shaped cells undergoing TGF-β-induced EMT. A549 cells were incubated with the presence of TGF-β and/or SM for 48 h followed by the evaluation of cellular morphology by phase contrast microscopy and the calculation of spindle-like cell morphology score, according to the formula (1), using ImageJ tool. (**B**) SM effectively decreases wound healing rate of TGF-β-stimulated A549 cells. The scratched cell monolayers were incubated with the presence or absence of TGF-β (50 ng/mL) and SM (0.5 µM) for 24 h and 48 h after that the analysis of wound closure capacity was carried out using phase contrast microscopy data. (**C**) SM inhibits TGF-β-stimulated motility of A549 cells. The cells were seeded in the upper chamber of cell invasion/migration (CIM)-Plate and treated with TGF-β (50 ng/mL) and/or SM (0.5 µM) for 48 h. Fetal bovine serum (FBS)-mediated migration of the cells to the lower chamber was analyzed using the xCELLigence Real-Time Cell Analyzer Dual Plates (RTCA DP) system. (**D**) SM inhibits TGF-β-stimulated invasion of A549 cells. The cells were seeded in the upper chamber of CIM-Plate, the bottom of which was covered by Matrigel, and treated with TGF-β (50 ng/mL) and/or SM (0.5 µM) for 48 h. Invasion of the cells to lower chamber, containing 10% FBS, was analyzed by xCELLigence RTCA DP instrument.

**Figure 6 molecules-25-05925-f006:**
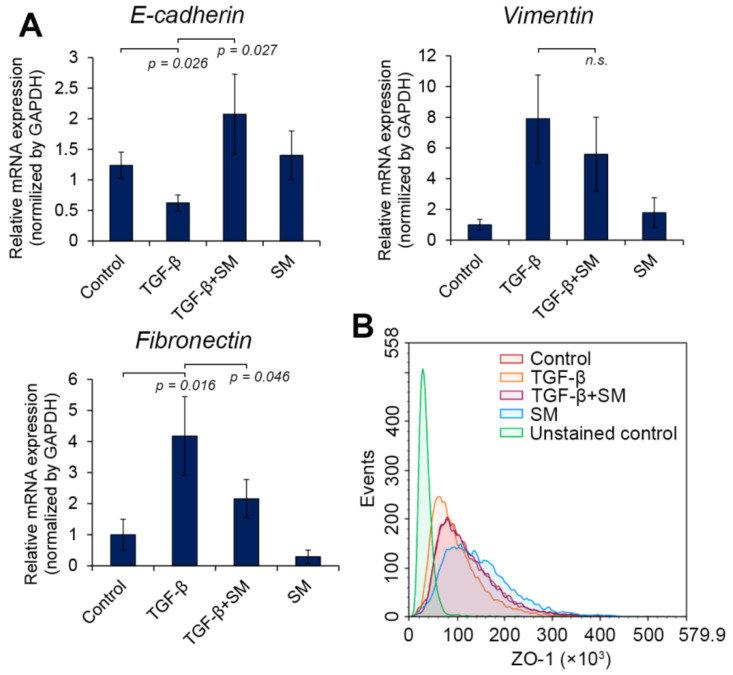
SM switches expression of EMT-associated molecular markers to epithelial ones in TGF-β stimulated A549 cells. The cells were treated with TGF-β (50 ng/mL) and/or SM (0.5 µM) for 48 h followed by evaluation of E-cadherin, vimentin and fibronectin mRNA (**A**) and ZO-1 protein (**B**) expression using RT-PCR and flow cytometry, respectively. Relative expression levels in A were shown as mean ± standard deviation from two independent experiments performed in tetraplicate.

**Figure 7 molecules-25-05925-f007:**
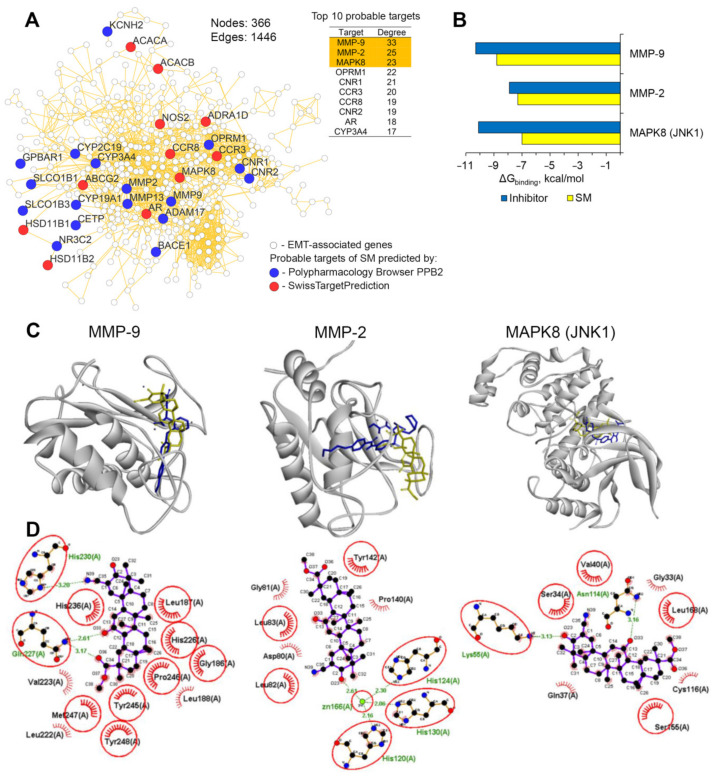
Network pharmacology revealed c-Jun N-terminal protein kinase 1 (JNK1) and matrix metalloproteinases 2 and 9 (MMP-2/-9) as probable primary targets of SM, associated with its anti-EMT activity. (**A**) The gene association network consisted from EMT-related regulome and predicted SM’s protein targets was reconstructed using the STRING database (confidence score > 0.7) in Cytoscape. The top 10 interconnected probable targets of SM are listed in the table. Degree is the number of interconnections of analyzed node into the regulome. (**B**) The binding energies (ΔG_binding_) of SM and the known inhibitors with the top 3 revealed EMT-associated probable targets of the triterpenoid calculated using Autodock Vina. (**C**,**D**) The mode of binding of SM to MMP-9, MMP-2 and JNK1. (**C**) 3D representation of docked poses of SM, superimposed on inhibitor bound structures of mentioned proteins was drawn by BIOVIA Discovery Studio. (**D**) 2D representation of docked poses of SM in MMP-9, MMP-2 and JNK1, created by LigPlot+. The green lines and red combs represent hydrogen bonds and non-bonding contacts, respectively. The amino acid residues, being commonly involved in the interaction networks of SM and the corresponding inhibitors are highlighted in red circles.

**Figure 8 molecules-25-05925-f008:**
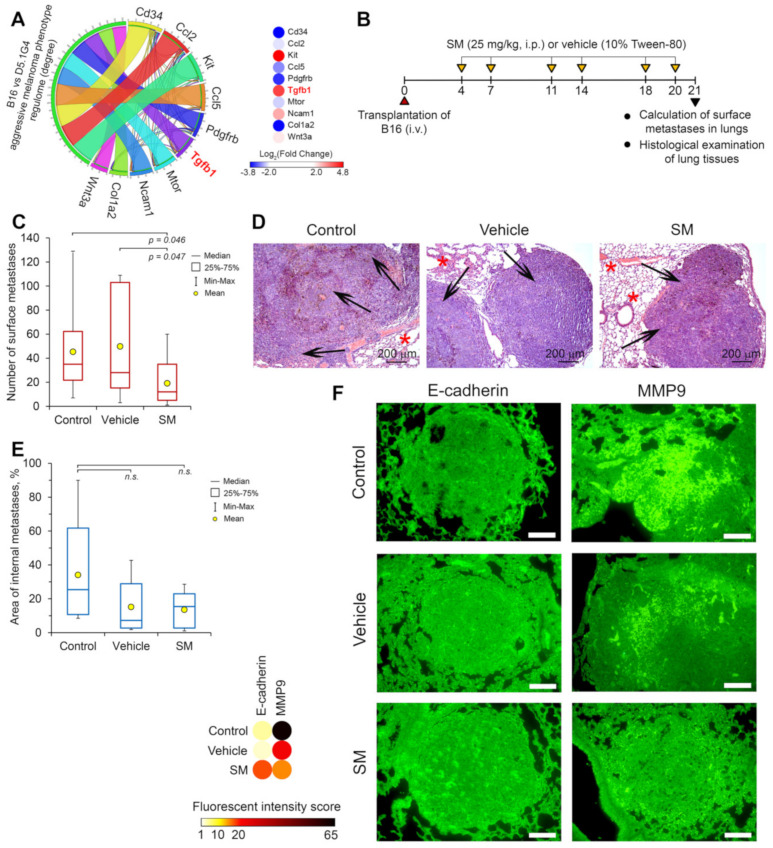
SM suppresses the metastatic growth of B16 melanoma in vivo. (**A**) TGF-β plays a key role in regulation of high tumorigenicity of B16 melanoma cells. The whole genome expression profiles of highly aggressive B16 and poorly tumorigenic D5.1G4 murine melanoma cells were compared by re-analysis of GSE69908 dataset using GEO2R tool. The left panel shows the top 10 master regulators, which control the growth and metastasis of B16 melanoma and are the most associated with analyzed regulome. Degree—the number of interconnections between master regulators and neighbors in gene regulatory network reconstructed from obtained differentially expressed genes (|Log_2_(Fold Change)|>2, *p* < 0.05) using the Search Tool for the Retrieval of Interacting Genes (STRING) database. The right panel shows the changing of expression of revealed master regulators between B16 and D5.1G4 cells (red and blue represent up- and down-regulation, respectively). (**B**) The scheme of the animal study. (**C**) The number of surface metastases in the lungs of B16 melanoma-bearing mice. (**D**,**E**) Histological images (**D**) and areas of internal metastases (**E**) of B16 melanoma in the lungs of tumor-bearing mice without treatment and after SM or vehicle administration. Hematoxylin and eosin staining. Original magnification ×100. The black arrows and red asterisks indicate lung metastasis foci and normal lung tissue, respectively. (**F**) SM suppresses the expression of EMT-associated markers on the model of B16 melanoma in vivo. Representative images of immunohistochemical staining of lung metastases with anti-E-cadherin and anti-MMP9 primary antibodies with subsequent incubation with secondary Alexa Fluor^®^ 488-conjugated antibodies. Scale bar corresponds to magnification ×200. The intensity of green fluorescence, corresponding to the expression of E-cadherin and MMP-9, was calculated according to the formula (2) for each image after subtraction of background using ImageJ software and visualized as a heatmap using Morpheus tool.

**Table 1 molecules-25-05925-t001:** The parameters of molecular docking simulations.

Protein	PDB ID	Center			Size		
		x	y	z	x	y	z
JNK1	4AWI	23.09	14.05	30.73	18	22	14
MMP-2	1HOV	6.91	18.73	22.84	14	20	18
MMP-9	1GKC	1.26	50	19.72	20	14	14

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
