# Peer review of "Cyano Enone-Bearing Triterpenoid Soloxolone Methyl Inhibits Epithelial-Mesenchymal Transition of Human Lung Adenocarcinoma Cells In Vitro and Metastasis of Murine Melanoma In Vivo"

_molecules, 2020, doi:10.3390/molecules25245925_

Round 1

Reviewer 1 Report

The author aims to better understand the antitumor activity of cyano enone-bearing semisynthetic compounds and reveal that SM is a promising anti-metastatic drug candidate.

  1. Please show p-value in Fig2.
  2. In Fig5C, treatment with SM only cannot reduce the migrating ability of A549 cells. How does this result reflect the data of Fig3B.
  3. In Fig8C and Fig8E, the image resolution needs to be improved, and clearly indicate the image signal to be observed and compared.
  4. The author proves that SM has the ability to inhibit EMT and metastasis in A549 cell model, mouse B16 melanoma metastasis model and computer model analysis. But the relationship between each other is not clearly linked and explained. For example, in the A549 cells, SM can inhibit the EMT activity induced by TGF-B, but in the metastatic model of murine B16 melanoma, it is not mentioned whether it is related to TGF-B and how high is the activity of TGF-B in B16?

Author Response

Dear Reviewer #1,

We are sincerely thankful to you for your deep analysis of our manuscript and highly valuable remarks. We revised the manuscript according to your comments and, please, let us respond to your questions.

1) Please show p-value in Fig2.

Authors: Corrected.

2) In Fig5C, treatment with SM only cannot reduce the migrating ability of A549 cells. How does this result reflect the data of Fig3B.

Authors: Corrected. Indeed, our results showed that soloxolone methyl (SM) statistically significantly inhibited motility of A549 in wound healing assay (Figure 3B), however, did not affect transwell migration of the cells in CIM-Plate. We suppose that this dissimilarity can be explained by the presence of 10% FBS in the lower chamber of CIM-Plate used as chemoattractant. In the case of wound healing assay, the triterpenoid effectively suppressed basal migratory activity of malignant cells (Figure 3B); however, the capacity of this effect is not enough to block cellular motility in the presence of additional migratory stimulus (FBS) (Figure 5B). Actually, we have already given this supposition about probable causes of observed discrepancy in Discussion section, however, in order to describe this more clearly, some corrections were introduced into mentioned paragraph (please, see lines 271-275).

3) In Fig8C and Fig8E, the image resolution needs to be improved, and clearly indicate the image signal to be observed and compared.

Authors: Corrected. The resolution of images in Figure 8D and Figure 8F was increased up to 1000 dpi. Additionally, the raw data obtained from light and fluorescent microscopy was also added to Supplementary materials (Please, see Figures S5-S12; lines 447-448 and 468-469). Besides this, in the case of Figure 8D, normal lung tissue was additionally marked by red asterisks in order to help readers to distinguish metastatic foci from non-transforming tissues (please, see line 434). In the case of fluorescent microscopy data, additional phrase was added to the description of the Figure 8F, describing that green fluorescent signals in the photos after subtraction of background correspond to the expression of E-cadherin and MMP-9 (please, see lines 439-441).

4) The author proves that SM has the ability to inhibit EMT and metastasis in A549 cell model, mouse B16 melanoma metastasis model and computer model analysis. But the relationship between each other is not clearly linked and explained. For example, in the A549 cells, SM can inhibit the EMT activity induced by TGF-B, but in the metastatic model of murine B16 melanoma, it is not mentioned whether it is related to TGF-B and how high is the activity of TGF-B in B16?

Authors: Corrected. Thank you for this highly valuable remark. Indeed, we poorly linked two parts of our work, describing the effects of SM on TGF-β-driven EMT in vitro and metastasis of B16 melanoma in vivo, between each other. In order to show the key regulatory role of TGF-β in aggressiveness of B16 melanoma cells, additional bioinformatic analysis of cDNA microarray data was performed. As a result of comparison of whole genome expression profile of highly aggressive B16 cells with that of poorly tumorigenic D5.1G4 murine melanoma cells, a set of differentially expressed genes (DEGs), involved in regulation of melanoma progression and metastasis, were identified. Further reconstruction of gene regulatory network from DEGs (Please, see Figure S3) and its analysis revealed top 10 key nodes – the genes, being the most interconnected with neighbors within the regulome and, therefore, playing an important master regulatory role in aggressiveness of B16 cells. We found that TGF-β (encoded by Tgfb1 gene) is involved in the list of identified key nodes (Please, see Figure 8A, left panel) and, moreover, the expression of TGF-β was ~17-fold up-regulated in B16 cells compared to its poorly tumorigenic D5.1G4 counterpart (Please, see Figure 8A, right panel). Furthermore, some key published works, describing the involvement of TGF-β in regulation of metastasis of B16 melanoma, were also considered in the chapter 2.3.5 (please, see lines 391-415).

We hope that this version of the manuscript will be acceptable for publication.

Thank you!

Reviewer 2 Report

The study by Markov et al entitled “Cyano Enone-Bearing Triterpenoid Soloxolone Methyl Inhibits Epithelial-Mesenchymal Transition of Human Lung Adenocarcinoma Cells in Vitro and Metastasis of Murine Melanoma in Vivo” has been reviewed. The study evaluates the effects of 18βH-glycyrrhetinic acid derivative soloxolone methyl (SM) used in non-toxic dosage on the processes associated with the metastatic potential of tumour cells.

The manuscript is clear and well written, as I believe the results can be of benefit to the research community, however, I have a comment regarding the preliminary toxicity analysis of the compounds.

Since the normal cell models roughly reflect toxicity profiles and are needed to better detect the therapeutic/toxic profile of a compound, why did not the author's test cytotoxicity of cyano enone-containing semisynthetic triterpenoids on normal cells such as an epithelial cell line? Please clarify this point.

So, overall I am positive about this study.

Author Response

Dear Reviewer #2,

We sincerely thank you for careful analysis of our manuscript and for very useful comment. We revised the manuscript and, please, let us respond to your remark.

Since the normal cell models roughly reflect toxicity profiles and are needed to better detect the therapeutic/toxic profile of a compound, why did not the author's test cytotoxicity of cyano enone-containing semisynthetic triterpenoids on normal cells such as an epithelial cell line? Please clarify this point.

Authors: Corrected. Indeed, the cytotoxicity of soloxolone methyl (SM) in non-malignant cells has not yet been published by our group. In order to evaluate it, the effect of SM on viability of non-transformed human gingival fibroblasts was studied. We showed that SM displays moderate selectivity toward tumor cells: IC50 of SM in fibroblasts was 5.9±0.4 µM after 24 h of treatment (Please, see Figure S1 and lines 159-163), whereas this value in A549 cells in similar treatment regimen was 2.2±0.1 µM. Anyway, used concentration of SM for evaluation of its anti-EMT activity (0.5 µM) was non-toxic to both malignant and non-malignant cells. Besides this, sixfold intraperitoneal injections of SM at dosage of 25 mg/kg did not cause the loss of weight of experimental animals and, therefore, was non-toxic in vivo.

We hope that this version of the manuscript will be acceptable for publication.

Thank you very much!

Round 2

Reviewer 1 Report

The author has clearly responded to previous suggestions.

In the comparative experiment analysis, the calculation and comparison of each group of experiments are based on the control group as the reference value, so the value of each control group should be 1 or 100%. Why is the value of the E-cadherin mRNA expression control group in Figure 6A not 1?  Moreover, the relative values ​​of many control groups vary greatly, for example, as shown in Figure 3C.